# A Mass Balance of Nitrogen in a Large Lowland River (Elbe, Germany)

**Stephanie Ritz** *,† **and Helmut Fischer**

Federal Institute of Hydrology—BfG, Am Mainzer Tor 1, 56068 Koblenz, Germany; Helmut.Fischer@bafg.de
* Correspondence: Stephanie.Ritz@bfn.de
† Current address: Federal Agency of Nature Conservation—BfN, Konstantinstrasse 110, 53179 Bonn, Germany.

**Abstract:** Nitrogen (N) delivered by rivers causes severe eutrophication in many coastal waters, and its turnover and retention are therefore of major interest. We set up a mass balance along a 582 km river section of a large, N-rich lowland river to quantify N retention along this river segment and to identify the underlying processes. Our assessments are based on four Lagrangian sampling campaigns performed between 2011 and 2013. Water quality data served as a basis for calculations of N retention, while chlorophyll-a and zooplankton counts were used to quantify the respective primary and secondary transformations of dissolved inorganic N into biomass. The mass balance revealed an average N retention of 17 mg N $m^{-2}\,h^{-1}$ for both nitrate N ($NO_3$–N) and total N (TN). Stoichiometric estimates of the assimilative N uptake revealed that, although $NO_3$–N retention was associated with high phytoplankton assimilation, only a maximum of 53% of $NO_3$–N retention could be attributed to net algal assimilation. The high TN retention rates in turn were most probably caused by a combination of seston deposition and denitrification. The studied river segment acts as a TN sink by retaining almost 30% of the TN inputs, which shows that large rivers can contribute considerably to N retention during downstream transport.

**Keywords:** nitrogen retention; mass balance; large rivers; denitrification; phytoplankton assimilation; Lagrangian sampling

## 1. Introduction

In the past centuries, human activities massively altered the natural biogeochemical cycle of N. In particular, the development of industrial N fixation with the Haber–Bosch process in the early 20th century, which allowed the large-scale production of fertilizer, led to a dramatic, continuous accumulation of reactive N in our ecosystems [1,2]. In-stream processes can substantially alter the reactive N load transported from catchments to coastal areas. Assimilation, for instance, reduces the dissolved N fraction from the river water by conversion to organic matter, and deposition of N compounds in the riverbed alters the total N (TN) pool during downstream transport. However, due to possible re-mineralization and re-suspension, reactive N principally remains in the ecosystem and therefore only contributes to a temporal retention of N. Denitrification, in contrast, leads to a permanent removal of reactive N from river water by gaseous losses of $N_2$ (and small amounts of $N_2O$) and is therefore of particular interest in mitigating N pollution.

However, to date the majority of empirical data on N removal and denitrification arise from studies on small streams, which limits our present understanding on N removal across whole river networks. Concerning the relative role of small vs. large rivers in N retention, the existing studies offer varying conclusions. Since N retention is usually linked to biogeochemical processes that are associated with streambed sediments (e.g., denitrification) [3], it is often emphasized that small headwater streams are especially effective in N removal because of their lower ratio of water column to benthic surface

area and because of their longer surface water residence time per unit channel length [4,5]. Other studies emphasize high N retention rates in large rivers due to cumulative effects along a river network: Large rivers transport higher N loads, since they receive additional N inputs from upper sections [6,7]. Furthermore, large rivers are less shaded, leading to a higher autotrophic N uptake and consequently to a higher supply of easily degradable autochthonous carbon [8] that serves as a substrate for N removal via denitrification. Therefore, some river network models suggest that although a larger proportion of the N inputs is retained in smaller streams, by absolute amounts, larger rivers retain more mass [9,10].

However, there are still methodological difficulties that arise when N retention should be measured on an ecosystem scale. Methods that measure biogeochemical processes in chambers or incubated samples rarely provide system-level quantifications of N retention because of enclosure effects and the spatial heterogeneity of the substrate [11,12]. Furthermore, especially in large rivers, biogeochemical activity may occur well below the sediment–water interface [13,14], where metabolic processes cannot easily be mapped, and in situ additions of stable isotope tracer such as $^{15}$N in sufficient amounts would yield high costs and are limited because of incomplete mixing. Therefore, especially large rivers have rarely been studied, and there remains much uncertainty about the actual N retention rates in these systems.

In contrast, the mass balance approach estimates net N retention as the difference between input and output fluxes and therefore integrates N processed in different sediment types and depths and on larger scales and does not rely on the addition of external tracers. Importantly, for a successful application of this method, the study site has to be selected carefully. All significant inputs and outputs to the investigated have to be well constrained, while reach length needs to be long enough to obtain a measurable change in N concentrations [15,16]. Therefore, only few mass balance studies have been performed on larger rivers with discharge >50 m$^3$ s$^{-1}$ (e.g., [17]).

Here, a mass balance was set up to quantify the retention of both nitrate N (NO$_3$–N) and TN along a 582 km-long section of an eighth order lowland river, the Elbe. For this river segment, there exists evidence for a high NO$_3$–N turnover [18] as well as high denitrification rates [13,19], but independent measurements are needed to verify these results. The main objective of this study is to provide in situ quantifications of N retention in the Elbe on an ecosystem scale and thereby improve the understanding of the extent to which large lowland rivers can act as an N sink. However, the Elbe is a highly eutrophic river with pronounced phytoplankton blooms during spring and summer [20], and assimilation has a large impact on NO$_3$–N retention [18,21]. To clarify whether the quantified retention is temporal (based on assimilation or deposition) or permanent (based on denitrification), we provide estimates about the relative role of phytoplankton assimilation against other NO$_3$–N retention pathways by relating the production of chlorophyll-a (chla) and zooplankton to the concurrent retention of NO$_3$–N and TN. We hypothesize that, due to the high NO$_3$–N and autochthonous carbon concentrations, not only assimilation of NO$_3$–N by phytoplankton, but also other processes (i.e., denitrification or deposition of previously assimilated NO$_3$–N) account for a considerable proportion of the N budget in the Elbe. This should result not only in a high NO$_3$–N retention but also in a high retention of TN. While we expect the TN retention rate (retention per unit area and time) to be high, with reference to the large N inputs into the investigated segment and into the river network models that state a lower efficiency of TN retention in large rivers [4,5,8], we expect the proportion of TN that is retained within this reach (reach-specific N removal in %) to be comparatively low.

## 2. Materials and Methods

### 2.1. Study Site

With a catchment area of 148,268 km$^2$ and a total length of 1094 km, the Elbe is one of the largest rivers in Central Europe. The Elbe originates in the Krkonoše ("Giant") Mountains close to the Czech–Polish border and flows in a wide curve of 367 km through the Czech Republic. In this section, the Elbe is impounded by a total of 24 weirs. Downstream of Střekov (329 km from the source),

the river is free flowing over a distance of 622 km down to the Geesthacht weir, where the 142 km-long tidal segment begins [22,23]. In this study, the Elbe-km refer to the German navigation kilometers, defining the German–Czech border as Elbe-km 0 (Figure 1).

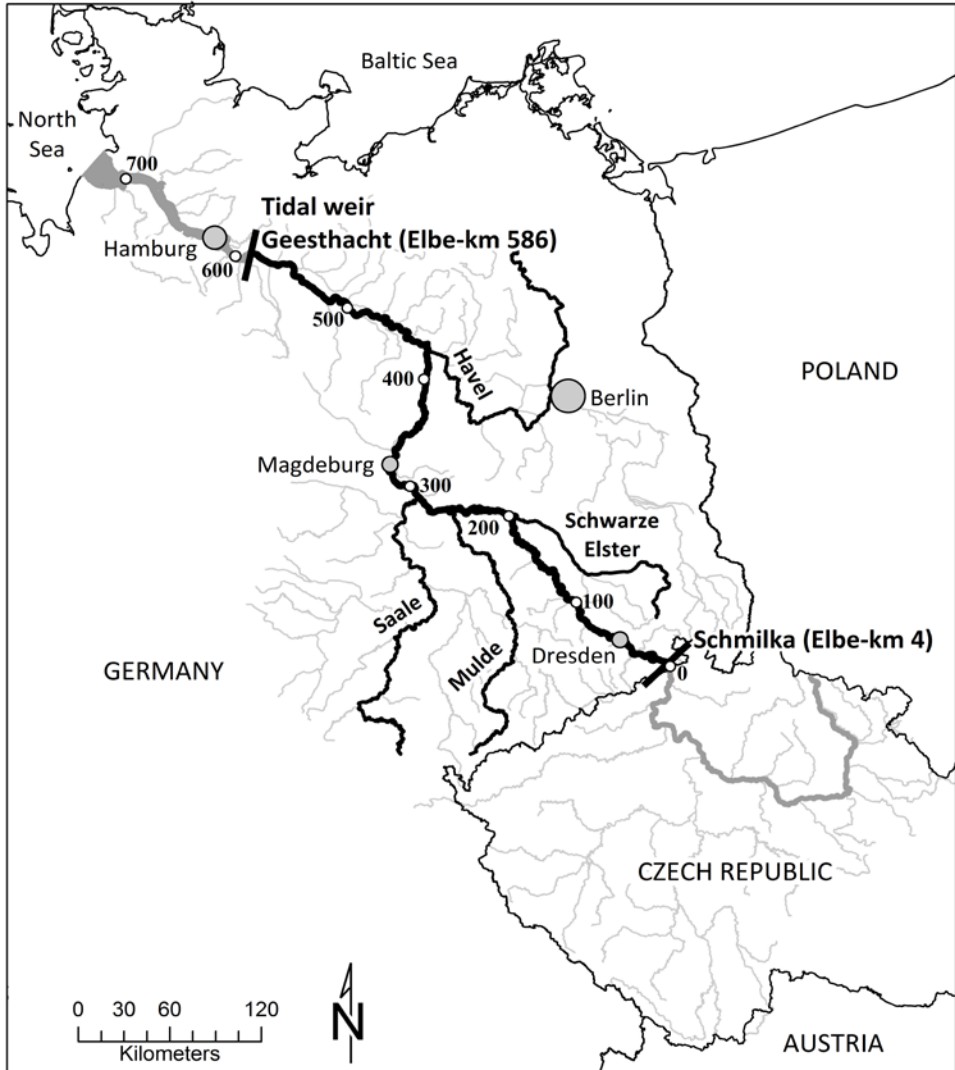

**Figure 1.** Catchment area of the Elbe. The black bars mark the boundaries of the study segment. White dots: Elbe-km according to the German navigation kilometers.

The studied segment starts close to the Czech–German border (Elbe-km 4) and ends upstream of the tidal weir Geesthacht at Elbe-km 586 (Figure 1). Since 90% of the surface water inputs within that segment derive from the upstream section and from four major tributaries (Schwarze Elster, Mulde, Saale, and Havel) that can easily be quantified by direct sampling, this section is an appropriate site for large-scaled mass balances. Groundwater inflow is not expected to considerably bias the mass balance, because the groundwater residence time in the catchment is long (~30 years [24]) and the direct exchange between groundwater and river water occurs on a relatively small scale (e.g., during flood events, infiltrating river water needs 48 days to be relocated 8 m away from the main channel [25]).

Around 25 million people live within the Elbe catchment. Land use is dominated by crop fields (51%), forests (29%), other agricultural uses (12%), and settlements and roads (7%). The dominance of agriculture is reflected in the high nutrient inputs; 73% of the TN inputs derive from diffuse sources [23]. Although nutrient concentrations greatly decreased after the reunification in 1989, in the downstream section of the Elbe, the annual means of TN and total P remain high, with concentrations

>3 mg N L$^{-1}$ and >0.1 mg P L$^{-1}$ (data refer to the monitoring station Schnackenburg at Elbe-km 475 for the years 1985–2012) [26]. Natural nutrient concentrations in the Elbe river system, based on background nutrient immissions, are estimated at <0.03 mg L$^{-1}$ of total phosphorus and <1 mg L$^{-1}$ of TN [27]. Due to the high N concentrations, the river is a significant source of N to the German bight, whose estuaries and coastal waters are classified as problem area with regard to eutrophication [28]. In the vegetation period (April–September), the chla concentration of the Elbe considerably increases longitudinally, albeit the tributaries tend to dilute the concentration at each river mouth [21]. On average, chla concentration increases by 8%–25% per flow day during downstream transport, with higher rates usually occurring in summer when discharge and water levels are low [29]. Besides the intense phytoplankton assimilation in the water column, also microbial activity in the streambed is high. The shifting sediments of the free-flowing river lead to the formation of subaqueous sand dunes, supporting an intense hydrologic exchange between the river water and sediments. Therefore, the riverbed is continuously supplied with nutrients and organic substrate (phytoplankton biomass), providing favorable conditions for nutrient and carbon turnover [13,30]. On the basis of measurements of microbial metabolism, it was estimated that 1.4% of the transported organic carbon per river km is degraded. This carbon turnover length is an order of magnitude shorter than would be expected by a regression of discharge versus turnover length for a set of 26 streams and rivers from various biomes [13].

## 2.2. Sampling and Laboratory Analyses

Four independent sampling campaigns were performed along the free-flowing German part of the Elbe between Schmilka (Elbe-km 4) and Geesthacht (Elbe-km 586). Three sampling campaigns took place during summer (8–15 August 2011, 25 July–2 August 2012, 12–19 August 2013), and one was conducted in spring (6–12 May 2012). A Lagrangian sampling scheme was applied by which a specific object (here, the parcel of water moving downstream) is tracked through space and time rather than at a specific location through time (Eulerian approach). This is more useful to identify in-stream processes altering the chemical, physical, and biological properties during downstream transport [31,32]. For each campaign, the flow time of the water was calculated with the 1-dimensional hydrodynamic model HYDRAX [33] to minimize time-related biases in the calculations of the flow-time-concordant sampling sites and the input and output fluxes for the investigated river segment. The model uses real-time discharge data from the upstream section and the main tributaries as input parameters and includes cross-sectional profiles of the Elbe for every 500 m as well as river slope and bed roughness for the entire 580 km-long river segment. HYDRAX is used regularly as a basis for water quality modeling of the Elbe. Validations of the model showed high Nash–Sutcliffe efficiencies and low relative errors (e.g., [34]). All samples were taken according to the calculated flow time of a distinct water parcel of the Elbe with a 10 L bucket from the fully mixed water column. Elbe water was sampled twice a day (6:00–9:00 and 17:00–18:00) and in triplicate (one sample from the middle, one from the left side, and one from the right side) by launching a small motor boat at the calculated time and location. Samples were prepared immediately in a mobile laboratory (laboratory bus) providing all the necessary instruments, measuring, and filtering devices. The four major tributaries (Schwarze Elster, Mulde, Saale, Havel) were sampled close to their inflow into the Elbe. In 2011, additional water samples were taken from the seven major wastewater treatment plants (WWTP) in Dresden, Meißen, Riesa, Wittenberg, Dessau, Schönebeck, and Magdeburg directly at their outflow. The sampling of the tributaries and the WWTPs was done within a 12 h time span with respect to the calculated time of each inflow into the Elbe. Because the share of the WWTP of the total loads of the Elbe was less than 3% for all measured parameters and because concentrations at these outflows were similar to our results in a previous study [18], the WWTP inputs measured in 2011 were also used for the other campaigns.

For zooplankton, 10–20 L of river water was filtered through a plankton net (mesh size 55 μm), concentrated in a 50 mL bottle and fixed with formaldehyde (4%). The water samples were processed and fixed directly after sampling in the laboratory bus and stored for subsequent analyses. Concentrations

of the dissolved nutrients as well as TN, chla, suspended substances (seston), and chloride ($Cl^-$) were determined using German Standard Methods [35]. The difference between the inputs of the dissolved inorganic N compounds $NO_3$–N, nitrite N ($NO_2$–N), and ammonium N ($NH_4$–N) and the TN input were termed "organic N", because this fraction should basically consist of algal, detrital, and dissolved organic N. Zooplankton was counted, and taxa were identified on a phylum or subphylum level.

### 2.3. Mass Balance Calculation of $NO_3$–N, TN, and Chlorophyll-a

The mass flux or load of a solute past a measurement station is defined as the product of the solute concentration (expressed in mass per volume) and the water discharge (volume per time). To calculate the mass balance, the mass flux at the output of the studied segment was compared to the sum of all measured and estimated inputs. Therefore, concentration measurements from the upstream and downstream location of the Elbe (Elbe-km 4 and 584, respectively) as well as the samples taken from the main tributaries (Schwarze Elster, Mulde, Saale and Havel) were used together with the corresponding discharge data at the gauging stations Schöna (Elbe at Elbe-km 2.1), Löben (Schwarze Elster, 21.6 km upstream of confluence), Bad Düben (Mulde, 68.1 km upstream of confluence), Calbe-Grizehne (Saale, 17.6 km upstream of confluence), Havelberg (Havel, 20.5 km upstream of confluence), and Neu Darchau (Elbe at Elbe-km 536.4) to calculate the respective load. The discharge data were provided by the WSV (Federal Waterways and Shipping Administration), except for Schwarze Elster and Mulde, whose data were provided by the regional authorities LfULG (Sächsisches Landesamt für Umwelt Landwirtschaft und Geologie) and LHW (Landesbetrieb für Hochwasserschutz und Wasserwirtschaft Sachsen-Anhalt), respectively. Furthermore, the NITROLIMIT database (unpublished monitoring data of the regional environmental authorities compiled within the scope of the research project NITROLIMIT) was analyzed with regard to the remaining, smaller tributaries directly discharging into the Elbe (surface runoff). The database covers a maximum time span from 2004 to 2011, depending on the respective water body. Because nutrient inputs did not change significantly between the years 2004 and 2013 [26], the time lag between the monitoring data covered in the database and the empirical data collected during the sampling was not expected to bias the mass balance calculations considerably. To address the uncertainties associated with seasonal fluctuations, for each tributary listed in the database, the respective water quality and discharge data were averaged for the corresponding sampling season (3-month average containing the month of the sampling campaign ± one month). These additional data improved the database for the mass balance by another 11% in terms of the covered catchment. The whole dataset (empirical measurements and database analysis) resulted in a total coverage of 94% of the contributing catchment area in which the catchment area at Elbe-km 586 (=135,000 $km^2$) was set as 100% (Figure 2). Additionally, the seven major WWTP sampled in 2011 were included into the mass balance. Further direct inputs were not explicitly incorporated in the mass balance calculations but were assessed from water balances and $Cl^-$ budgets that were set up for each campaign.

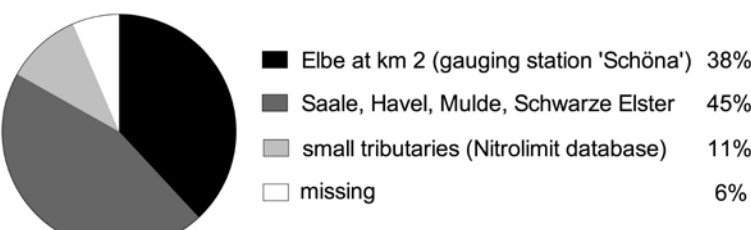

**Figure 2.** Shares of the sub-basins of the total catchment area of the studied river section. The drainage area of the Elbe at Geesthacht (Elbe-km 586) is set as 100%.

The retained mass flux (expressed as mass per unit time, $g\ s^{-1}$), that is directly calculated from the mass balance, not only reflects the average retention rate per unit area and time, but also is a function of the total water travel time (here, the total water travel time within the 582 km river stretch) and the wetted sediment surface that is overflown (here, the river bed of the 582 km-long river stretch).

Assuming the same retention rate per area and time, the retained mass flux will be higher in systems with longer water travel times and more surface area per volume of water. In order to achieve a broader comparability, we additionally calculated the retention rate per unit area and time (mg m$^{-2}$ h$^{-1}$) by relating the retained mass flux to the respective river area and water travel time that prevailed during each campaign. The river area and water travel time were calculated for each campaign using the hydrodynamic model HYDRAX.

The uncertainties of the mass balance computations were determined according to the principles of Gaussian error propagation. The uncertainty of each mass input or output (load) is given as:

$$\delta M_i = \sqrt{(\delta C_i/C_i)^2 + (\delta Q_i/Q_i)^2} \times (C_i \times Q_i) \tag{1}$$

with

$\delta$ = uncertainty in associated term
$M_i$ = mass input or output at a given sampling site $i$
$C_i$ = concentration at a given sampling site $i$
$Q_i$ = discharge at a given sampling site $i$

The uncertainty in concentration was defined as the analytical error for each laboratory test that was ±10% for TN and NO$_3$–N and ±0.68 mg L$^{-1}$ for Cl$^-$. For chla, the standard deviation of the replicates samples was used. The uncertainty in discharge was assumed to be ≤5% (pers. communication W. Wiechmann, BfG). The uncertainty in the loss (or gain) of a substance is given as:

$$\delta T_S = \sqrt{\sum (\delta M_i)^2} \tag{2}$$

with

$\delta$ = uncertainty in associated term
$T_S$ = net loss or gain of the substance $S$
$M_i$ = mass input or output at a given sampling site $i$

### 2.4. Calculations on Algal N Uptake and Zooplankton Grazing

The increase in chla was used as a surrogate parameter for phytoplankton biomass accrual and to estimate the corresponding transformation of NO$_3$–N to algal N. Therefore, algal biomass was calculated with a C/chla conversion factor of 25 (mass ratio), and NO$_3$–N uptake was estimated using the Redfield ratio (C/N = 6.6; molar ratio). The C/chla ratio was based on the total average of phytoplankton biovolume measurements that were additionally performed for each campaign [36] combined with a C/biovolume relationship derived from freshwater algae [37]. The zooplankton abundances were used to correct the net chla accrual and the corresponding net N assimilation for the impact of zooplankton grazing. Clearance rates of zooplankton (the volume of water cleared of food per unit time) can vary depending on the food concentrations in the water column and the zooplankton species. Rotifers are the most abundant zooplankton group in the Elbe, and during the vegetation period, particulate organic carbon in the water column of the Elbe ranges between 2 and 9 mg L$^{-1}$ [18,38]. For this food concentration, clearance rates for rotifer grazing should not exceed 5 μL individual$^{-1}$ h$^{-1}$ (μL Ind.$^{-1}$ h$^{-1}$) [39]. This rate was applied to the chla concentrations and to the rotifer abundances measured each evening, extrapolated to 24 h, load-weighted with the respective discharge of each day, and summed up to obtain the total amount of ingested chla for each campaign:

$$total\ ingested\ chla = \sum_i \left( CR \times ZooAb_i \times Chla_i \times \frac{10^6\,\mu L}{1L} \times \frac{24h}{day} \times Q_i \times \frac{1g}{10^6 mg} \right) \tag{3}$$

with

*total ingested chla* = total ingested amount of chlorophyll-a for each campaign (g s$^{-1}$)
*i* = specific sampling site
*CR* = clearance rate (µL Ind.$^{-1}$ h$^{-1}$)
*ZooAb$_i$* = zooplankton abundance in each evening (Ind. L$^{-1}$)
*Chla$_i$* = chlorophyll-a concentration in each evening (µg L$^{-1}$)
*Q$_i$* = average daily discharge (m$^3$ s$^{-1}$)

## 3. Results

### 3.1. Longitudinal Profiles of Chlorophyll-a, Dissolved Nutrients, TN, and Zooplankton

Chla concentrations increased along the studied river section during all campaigns (Figure 3). Initial chla concentration at Elbe-km 4 was the highest in May 2012 (63 µg L$^{-1}$), but in this campaign the increase stagnated after the first 100 km, and concentrations remained within a range between 100 and 123 µg L$^{-1}$ for the rest of the downstream transport. In contrast, during the other three campaigns in summer, chla concentrations increased almost continuously.

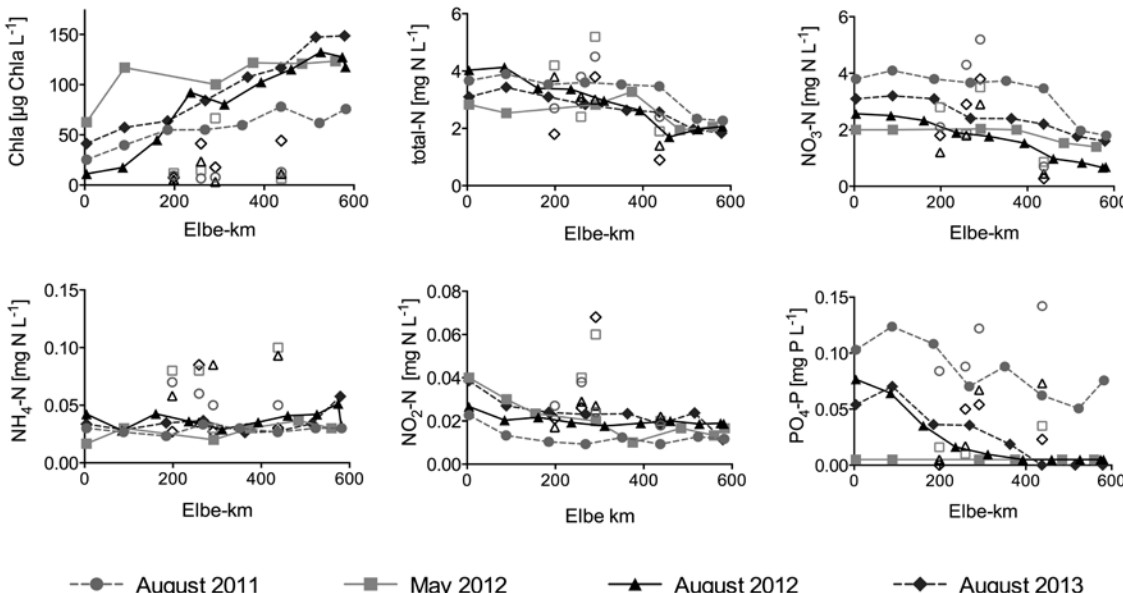

**Figure 3.** Concentrations of chlorophyll-a (chla), total N (TN), nitrate N (NO$_3$–N), ammonium N (NH$_4$–N), nitrite N (NO$_2$–N) and orthophosphate P (PO$_4$–P) in the different sampling campaigns. Filled symbols represent concentrations in the main river as average of the samples taken from middle, left, and right side at each evening sampling location. Empty symbols represent values of the major tributaries: Schwarze Elster (Elbe-km 198), Mulde (Elbe-km 259), Saale (Elbe-km 291), and Havel (Elbe-km 438).

The longitudinal chla increase was always accompanied by a considerable decline of NO$_3$–N, the dominating N compound in the Elbe (Figure 3). Nonetheless, with concentrations ranging between 2.0 and 3.8 mg N L$^{-1}$ at Elbe-km 4 and between 0.7 and 1.8 mg N L$^{-1}$ at the final sampling location upstream of the tidal weir, NO$_3$–N concentrations exceeded the demand for primary production throughout the river segment, since the threshold for limiting concentrations of dissolved N ranges between 0.03 and 0.1 mg L$^{-1}$ [40]. The Havel always carried considerably lower NO$_3$–N concentrations compared to the Elbe and thus had a diluting effect, whereas in the Saale, the concentrations were always higher. TN concentrations also decreased in the downstream direction (Figure 3). Initial values at Elbe-km 4 ranged between 4.0 mg N L$^{-1}$ and 2.9 mg N L$^{-1}$ and decreased to values between 3.7 mg N L$^{-1}$ and 2.3 mg N L$^{-1}$. As in the case of NO$_3$–N, TN concentrations in the Havel were

always lower compared to those in the Elbe, leading to a considerable dilution of TN concentrations in the Elbe. $NH_4$–N concentrations remained constantly low and ranged between 0.02 mg N $L^{-1}$ and 0.06 mg N $L^{-1}$. $NO_2$–N ranged between 0.01 mg N $L^{-1}$ and 0.04 mg N $L^{-1}$. In the tributaries, $NH_4$–N and $NO_2$–N concentrations were usually higher than in the Elbe. Orthophosphate P ($PO_4$–P) decreased longitudinally during the summer campaigns. Especially in August 2012 and 2013, the decrease was strong, and concentrations dropped below the limit of quantification of 0.01 mg P $L^{-1}$ during downstream transport. In May 2012, $PO_4$–P was always below 0.01 mg P $L^{-1}$, exerting a potentially limiting effect on phytoplankton growth in the downstream sections (cf. [40]). $PO_4$–P concentrations in the tributaries were generally slightly higher.

Zooplankton abundances increased in the downstream direction during every campaign (Figure 4). Initial zooplankton abundances at km 4 ranged between 11 and 46 Ind. $L^{-1}$ in the summer campaigns but were much higher in May 2012 (429 Ind. $L^{-1}$). In May 2012, a maximum value of 2135 Ind. $L^{-1}$ was detected at km 583. Rotifers clearly dominated the zooplankton composition in all campaigns.

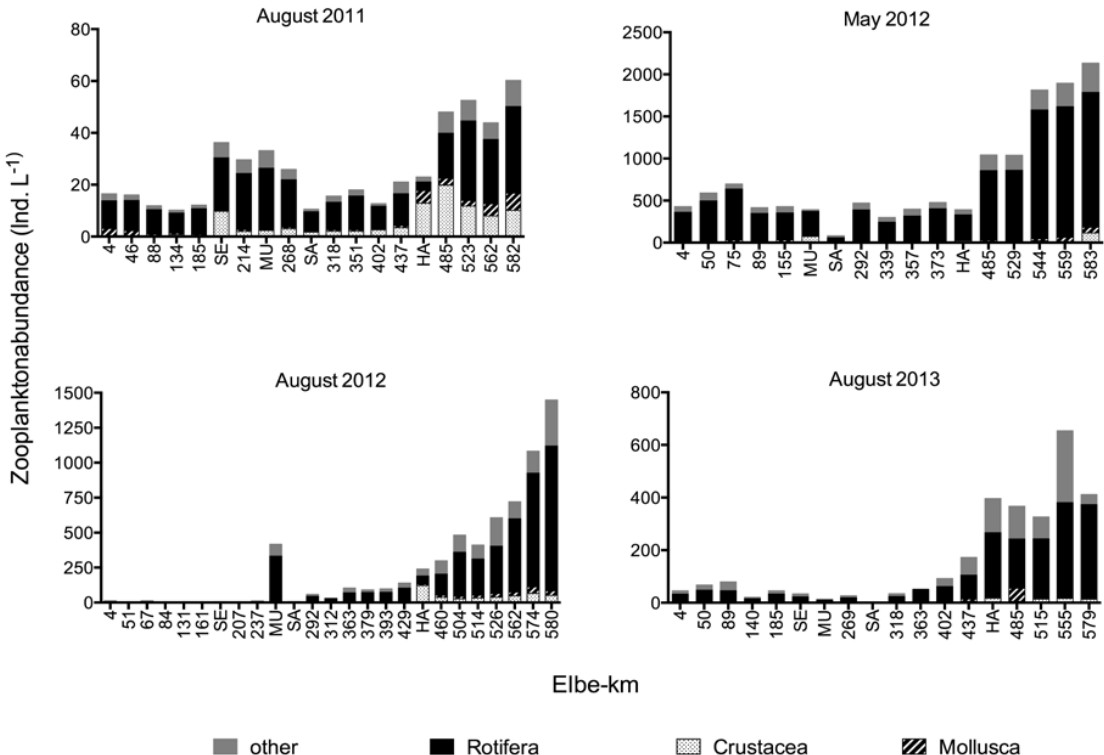

**Figure 4.** Abundances of the different zooplankton groups: Mollusca (larvae), Rotifera, Crustacea, and others during the four sampling campaigns. The numbers on the *x*-axis refer to the Elbe-km, the acronyms refer to the four major tributaries (SE = Schwarze Elster, MU = Mulde, SA = Saale, HA = Havel).

### 3.2. Mass Balance of $NO_3$–N, TN, and Chlorophyll-a

The total water travel time within the studied section ranged between 164 and 193 hours (Table 1). The amount of TN delivered to the studied river segment was between 1361 and 2030 g N $s^{-1}$ (Table 2). $NO_3$–N always contributed the major proportion of the TN inputs, ranging between 65% (May 2012) and 93% (August 2013). Organic N represented the second largest share of the N inputs and varied between 4.4% (August 2013) and 32% (May 2012). $NH_4$–N and $NO_2$–N accounted only for a minor proportion of the TN inputs, with contributions always below 2%.

**Table 1.** Overview of the hydraulic conditions during the sampling campaigns. *Q* = discharge; MQ = mean annual discharge for the years 1985–2015; travel time = water travel time from Schmilka (German Elbe-km 4) to "Geesthacht" (German Elbe-km 586) calculated with the hydrodynamic model HYDRAX. Discharge data derived from the gauging stations (Schöna, Neu Darchau, Löben, Bad Düben, Calbe-Grizehne, Havelberg) by the responsible authorities (WSV, LfULG and LHW).

| Date | Elbe-km 2 | Elbe-km 536 | Schwarze Elster | Mulde | Saale | Havel | Travel Time |
|---|---|---|---|---|---|---|---|
| | $Q$ (m$^3$ s$^{-1}$) | $Q$ (m$^3$ s$^{-1}$) | $Q$ (m$^3$ s$^{-1}$) | $Q$ (m$^3$ s$^{-1}$) | $Q$ (m$^3$ s$^{-1}$) | $Q$ (m$^3$ s$^{-1}$) | (h) |
| Aug. 2011 | 231 | 756 | 29 | 64 | 57 | 183 | 166 |
| May. 2012 | 283 | 588 | 11 | 38 | 69 | 101 | 164 |
| Aug. 2012 | 202 | 405 | 11 | 32 | 69 | 53 | 193 |
| Aug. 2013 | 260 | 515 | 15 | 35 | 80 | 49 | 169 |
| MQ | 304 | 683 | 18 | 65 | 115 | 109 | |

**Table 2.** Summary of the N fractions contributing to the TN input of the studied river segment. The values represent the sum of the load-weighted inputs of Elbe at km 4, the major and small tributaries, and the inputs by the wastewater treatment plants (WWTP) measured in 2011. The TN, NH$_4$–N, NO$_2$–N, and NO$_3$–N inputs refer to the measured values; organic–N was calculated as residual.

| Total Inputs of | August 2011 | | May 2012 | | August 2012 | | August 2013 | |
|---|---|---|---|---|---|---|---|---|
| | g s$^{-1}$ | % of TN | g s$^{-1}$ | % of TN | g s$^{-1}$ | % of TN | g s$^{-1}$ | % of TN |
| NH$_4$–N | 25.2 | 1.2 | 28.6 | 1.6 | 25.7 | 1.9 | 23.4 | 1.6 |
| NO$_2$–N | 15.2 | 0.7 | 20.8 | 1.2 | 10.5 | 0.8 | 14.9 | 1.0 |
| NO$_3$–N | 1745 | 86.0 | 1166 | 65.0 | 911 | 66.9 | 1333 | 92.9 |
| organic–N | 245 | 12.0 | 578 | 32.3 | 414 | 30.4 | 63.7 | 4.4 |
| TN | 2030 | 100 | 1793 | 100 | 1361 | 100 | 1435 | 100 |

Cl$^-$ concentrations and discharge data were used to calculate Cl$^-$ and water balances to detect unknown inputs along the studied segment. For spring 2012 and summer 2012 and 2013, all budgets were balanced within 10%, which was within the error calculated by the error propagation rule (Table 3). Only for August 2011, the Cl$^-$ budget showed a mismatch of +25%, and the water budget resulted in an apparent gain of +23%. For all campaigns, the total mass input of chla, NO$_3$–N, and TN differed clearly from the mass output, indicating that either net in-stream gains (output > input) or net in-stream losses (output < input) must have occurred (Table 3). Chla loads increased strongly along the studied segment, especially during the summer campaigns. Chla production was the highest in August 2013, accounting for 2.3 mg chla m$^{-2}$ h$^{-1}$ (60.4 g chla s$^{-1}$, 371%), and the lowest in spring 2012, with an in-stream production of 1.3 mg chla m$^{-2}$ h$^{-1}$ (34.4 g chla s$^{-1}$, 165%). During all campaigns, the chla increase was paralleled by distinct reduction of NO$_3$–N that ranged between −13.4 mg N m$^{-2}$ h$^{-1}$ (−343 g N s$^{-1}$, −29.4%) in May 2012 and −21.1 mg N m$^{-2}$ h$^{-1}$ (−635 g N s$^{-1}$, −69.8%) in August 2012. The TN retention rates were similar to the NO$_3$–N retention rates. The highest TN retention rate for the entire segment was detected in August 2012, with a total loss of −525.2 g N s$^{-1}$ (−38.6%). However, TN retention per unit area and time was the highest in August 2013, with −19.2 mg N m$^{-2}$ h$^{-1}$. The mass balances for NO$_2$–N and NH$_4$–N revealed no significant results (retention rates for both NO$_2$–N and NH$_4$–N always <30 g s$^{-1}$, with error values always >100 g s$^{-1}$, data not shown), because the differences between inputs and outputs were too low and are therefore not shown.

**Table 3.** Mass balance for the different sampling campaigns. Errors were calculated according to error propagation rules including errors in discharge (±5%) and concentration (analytical error). The area used to calculate net turnover rates refers to the wetted channel area and was calculated on the basis of the HYDRAX model output.

| Mass Balance for Each Campaign | chla (g s$^{-1}$) | NO$_3$–N (g s$^{-1}$) | TN (g s$^{-1}$) | Cl$^-$ (kg s$^{-1}$) | Discharge (m$^3$ s$^{-1}$) |
|---|---|---|---|---|---|
| **AUGUST 2011** | | | | | |
| Total input | 9.6 | 1745 | 2030 | 60.7 | 612.5 |
| Output Elbe-km 582 | 57.3 | 1361 | 1713 | 75.9 | 756 |
| Output–Input: (Δ load) | 47.7 ± 7.6 | −384 ± 120 | −316 ± 246 | 15.2 ± 4.3 | 143 ± 42.4 |
| (Δ%) | 499 ± 80.0 | −22 ± 6.9 | −15.6 ± 12.1 | 25.1 ± 7.1 | 23.4 ± 6.9 |
| Net turnover (mg m$^{-2}$ h$^{-1}$) | 1.8 ± 0.3 | −14.8 ± 4.6 | −12.2 ± 9.5 | | |
| **MAY 2012** | | | | | |
| Total input | 20.5 | 1166 | 1793 | 60.5 | 569.6 |
| Output Elbe-km 583 | 54.5 | 823 | 1352 | 64.2 | 587.8 |
| Output–Input: (Δ load) | 34.0 ± 9.4 | −343 ± 118 | −441 ± 188 | 3.7 ± 3.9 | 18.2 ± 34.8 |
| (Δ%) | 165 ± 45.8 | −29.4 ± 10.2 | −24.6 ± 10.5 | 6.1 ± 6.5 | 3.2 ± 6.1 |
| Net turnover (mg m$^{-2}$ h$^{-1}$) | 1.3 ± 0.4 | −13.4 ± 4.6 | −17.3 ± 7.3 | | |
| **JULY/AUGUST 2012** | | | | | |
| Total input | 4.2 | 910.5 | 1361 | 65.3 | 412.6 |
| Output Elbe-km 583 | 47.6 | 275.1 | 836.2 | 60.3 | 404.6 |
| Output–Input: (Δ load) | 43.4 ± 3.7 | −635 ± 90.7 | −525 ±146 | −5.0 ± 6.1 | 7.9 ± 26.8 |
| (Δ%) | 1036 ± 89.3 | −69.8 ± 10.0 | −38.6 ± 10.7 | −7.7 ± 9.3 | 1.9 ± 6.5 |
| Net turnover (mg m$^{-2}$ h$^{-1}$) | 1.4 ± 0.1 | −21.1 ± 3.0 | −17.4 ± 4.8 | | |
| **AUGUST 2013** | | | | | |
| Total input | 16.3 | 1333 | 1435 | 83.7 | 490 |
| Output Elbe-km 579 | 76.7 | 825.6 | 928 | 88.6 | 515 |
| Output–Input: (Δ load) | 60.4 ± 5.4 | −507 ± 159 | −506 ± 158 | 4.8 ± 7.8 | 25.4 ± 32.6 |
| (Δ%) | 371 ± 33.3 | −38.1 ± 11.9 | −35.3 ± 11.0 | 5.7 ± 9.3 | 5.1 ± 6.7 |
| Net turnover (mg m$^{-2}$ h$^{-1}$) | 2.3 ± 0.2 | −19.3 ± 6.0 | −19.2 ± 6.0 | | |

## 3.3. Estimates of Algal N Uptake and Zooplankton Grazing

Net phytoplankton assimilation accounted for net NO$_3$–N uptakes along the studied segment of 7.8, 5.6, 6.1, and 9.7 mg N m$^{-2}$ h$^{-1}$ in August 2011, May 2012, August 2012, and 2013, respectively, (Figure 5) and thus never represented the total amount of NO$_3$–N loss. Instead, its proportion ranged between 29% in August 2012 and 53% in August 2011. Because there was no significant (net) NH$_4$–N retention, no assimilative (net) uptake of NH$_4$–N was calculable.

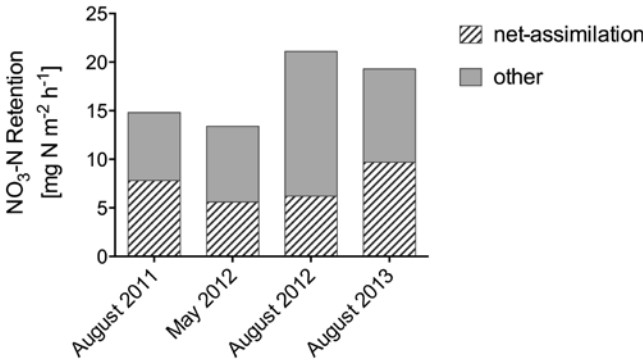

**Figure 5.** Calculated net assimilation share of NO$_3$–N retention for each sampling campaign. "other": NO$_3$–N retention that cannot be explained by net assimilation. The share of assimilation was calculated on the basis of chla gain. A C/chla ratio of 25 and the algal Redfield ratio were used as conversion factors. The area used to calculate turnover rates refers to the wetted channel area and was calculated on the basis of the HYDRAX model output.

The estimated impact of zooplankton grazing resulted in chla ingestions that ranged between 0.3 and 26.6 g chla s$^{-1}$ for August 2011 and May 2012, respectively (Table 4). Particularly in May 2012, zooplankton grazing affected chla development, with a corresponding N ingestion of 4.4 mg N m$^{-2}$ h$^{-1}$. This means that in the absence of zooplankton grazing, the calculated phytoplankton net assimilation of 5.6 mg N m$^{-2}$ h$^{-1}$ would have been 78% higher.

**Table 4.** Potentially grazed chla in each campaign. N transfer into zooplankton biomass was calculated using a C/chla ratio of 25 and the Redfield ratio. The area used to calculate N transformation rates refers to the wetted channel area and was calculated on the basis of the HYDRAX model output.

| Potentially Grazed Amounts | August 2011 | May 2012 | August 2012 | August 2013 |
|---|---|---|---|---|
| potentially grazed chla (g chla s$^{-1}$) | 0.3 | 26.6 | 9.2 | 5.9 |
| potentially grazed chla (% of net chla gain) | 0.7 | 78.4 | 21.2 | 9.8 |
| potential N transformation into zooplankton biomass (mg N m$^{-2}$ h$^{-1}$) | 0.1 | 4.4 | 1.3 | 1.0 |

## 4. Discussion

### 4.1. N Retention Rates and Realtive Role of Assimilation and Other Retention Processes

N retention averaged 17 mg N m$^{-2}$ h$^{-1}$ (460 g N s$^{-1}$) for both NO$_3$–N and TN and showed only little variation among the different campaigns of this study. Therefore, these values seem to be in a reliable range for N retention in the Elbe, at least for the vegetation period. Furthermore, for NO$_3$–N, an earlier individual sampling campaign carried out in July 2005 revealed a similar retention rate for the investigated Elbe segment (459 g N s$^{-1}$ [18]).

Assimilation leads to a temporary storage of dissolved inorganic N in biomass, and the phytoplankton growth in the Elbe coincided with distinct in-stream losses of NO$_3$–N. This mirrors the fact that during the growth season, phytoplankton dynamics have a major impact on the nutrient regime in the Elbe [21,34]. On the other hand, the share of NO$_3$–N retention attributable to net phytoplankton assimilation accounted only for a maximum of 53%. Under favorable conditions, metazoan grazers, particularly rotifers, can control the phytoplankton community in the downstream section of the Elbe [41] and other rivers [42,43]. Indeed, in May 2012, zooplankton grazing could have potentially biased the calculations on algal NO$_3$–N uptake to such an extent that almost 100% of the NO$_3$–N loss might have been caused by phytoplankton assimilation, with half of it subsequently incorporated in zooplankton biomass. During summer, zooplankton inoculation from the upstream, impounded river sections was lower, and grazing had only a minor impact.

Finally, there remains a considerable share of NO$_3$–N retention attributable to additional processes, i.e., deposition of previously assimilated NO$_3$–N and/or direct uptake of NO$_3$–N via microbial denitrification. This confirms our initial assumption that, besides the net assimilation of NO$_3$–N by phytoplankton, other processes such as denitrification or deposition (of previously assimilated NO$_3$–N) contribute to NO$_3$–N retention in the Elbe. The fraction of NO$_3$–N retention attributable to deposition and/or microbial denitrification ("NO$_3$–Nret—plankton Ngain") should be mirrored in the calculated TN retention, since both contribute to TN retention. However, for all campaigns, TN retention was even higher, which means that not only NO$_3$–N but also other N compounds were retained. During our samplings, N inputs other than NO$_3$–N ("TN—NO$_3$–N") ranged between 102 and 627 g N s$^{-1}$ (Table 2). Only a combination of partial losses of this "TN—NO$_3$–N" (via deposition) together with the fraction of NO$_3$–N attributable to deposition and/or microbial denitrification can explain the entire TN retention (Figure 6), particularly during the August campaigns. In spring 2012, deposition of the "TN—NO$_3$–N" compounds played a major role. Hence, besides net assimilation, other NO$_3$–N retention pathways such as deposition of previously assimilated NO$_3$–N and/or the direct uptake of NO$_3$–N by benthic biofilms contributed to the NO$_3$–N budget of the Elbe.

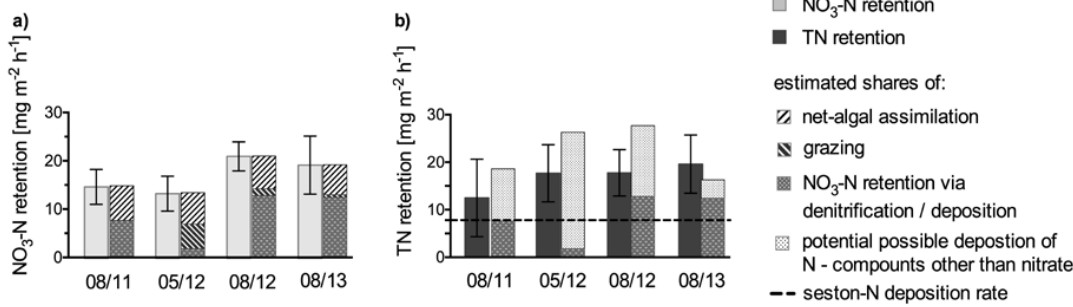

**Figure 6.** Calculated net assimilation share of $NO_3$–N retention for each sampling campaign. The share of assimilation was calculated on the basis of chla gain. A C/chla ratio of 25 and the algal Redfield ratio were used as conversion factors. The seston–N deposition rate was calculated after [44]. The area used to calculate the turnover rates refers to the wetted channel area and was calculated on the basis of the HYDRAX model output.

In the Elbe, sedimentation in hydrological dead zones (groin fields) and entrainment of particulate matter by advective transport into the hyporheic zone do contribute to the particle retention, and an average net seston retention of 62,000 t a$^{-1}$ between Elbe-km 455 and 523 has been estimated [44,45]. Assuming a mean N proportion of 1.5% in this settled material and an average river width of 200 m in this section, N deposition results to be 7.8 mg N m$^{-2}$ h$^{-1}$. Since TN retention found in this study exceeded these seston–N deposition rates, it seems most likely that both seston deposition and denitrification of water column $NO_3$ occurred. Furthermore, if deposition alone caused the entire TN retention, then large amounts of N would theoretically accumulate in the riverbed. However, no long-term accumulation of N occurs in rivers [46–48]. Instead, fresh and labile organic compounds (such as phytoplankton biomass) are usually consumed quickly and locally without downstream transport [48]. Finally, it can be concluded that, although the mass balance approach applied here cannot directly clarify to which extent deposition or denitrification caused TN retention, our results strongly indicate that a considerable part of N was removed by denitrification, either directly via turnover of water column $NO_3$–N, or indirectly via $NO_3$–N recycled from re-mineralized material. This is in line with open-channel measurements of denitrification that were simultaneously performed in the Elbe in summer 2011 and spring 2012 and revealed denitrification rates of 18 and 13 mg N m$^{-2}$ h$^{-1}$, respectively [19].

### 4.2. Methodological Limitations

The Lagrangian sampling scheme in combination with well-documented inflows from the catchment allowed a distinct separation between dilution effects and in-stream retention processes. The water and Cl$^-$ budgets that were used to trace the mixing with unknown inflows were in most cases below the error values calculated by the error propagation rule. Hence, the applied sampling scheme sufficiently accounted for all significant inputs along the studied river segment. The apparent gain of Cl$^-$ and water of 25% and 23%, respectively, that was detected in summer 2011, was presumably a consequence of the unusually high discharges of the Havel at that time [49]. Uncertainties in the Havel gauges at high discharges are well known, but no measured correction factor exists at the moment. Hence, for August 2011, the inputs were probably underestimated, which implies that the in-stream gain calculated by mass balance was possibly overestimated, while the in-stream losses were possibly underestimated.

The stoichiometric conversion factors that were used to relate the measured chla increase to $NO_3$–N uptake can vary. Particularly, C/chla ratios can change depending on environmental conditions such as irradiance, nutrient availability, and temperature [50]. In eutrophic environments, the conversion factor can vary between 16 and 83, with the majority of cases ranging between 27 and 67 [51]. This uncertainty in C/chla would lead to large uncertainties in the calculations of algal N uptake. The C/chla of 25 applied

in this study corresponds to the lower boundary of literature values. It provides a conservative estimate that is further supported by laboratory experiments performed with species dominating in the Elbe [52]. Additionally, for the gauging station at Magdeburg, an average C/chla ratio of 23 was reported [53], and similarly low C/chla ratios were found in the eutrophic rivers Meuse (Belgium) and Rideau (Canada) [54]. Furthermore, the exact grazing impact on chla increase is difficult to calculate. Clearance rates vary in the literature, even within the same species, depending on the ecological conditions (such as temperature, proportion of indigestible algae, etc.) or on the methodology [55]. However, May 2012 was the only case with zooplankton abundances high enough to affect phytoplankton growth and our estimates of assimilative N uptake. Benthic filter feeders such as bivalves are generally rare in the Elbe [56] and are therefore not expected to bias our calculations. Hence, despite the limitations imposed by the conversion factors used and their temporal dynamics, our results show that net assimilation alone could not account for the entire $NO_3$–N retention.

We acknowledge that only the net effects can be inferred by a mass balance, and the gross turnover rates can be higher. For instance, although $NH_4$–N concentrations remained on a constantly low level throughout the studied segment, assimilation of freshly produced $NH_4$–N was possible [57,58]. However, because no concentration (and load) differences were detected, it can be assumed that mineralization (the transformation of particulate organic N to $NH_4$–N) and assimilation (the transformation of $NH_4$–N to biomass) were balanced. This resulted in a zero effect in terms of net $NH_4$–N assimilation by phytoplankton, which therefore was assumed not to bias the calculations of the share of net assimilation in $NO_3$–N retention. The same holds true for the heterotrophic assimilation of $NO_3$–N, which may not have a large impact on $NO_3$–N retention because, in the Elbe, phytoplankton biomass exceeds the biomass of bacterioplankton and heterotrophic flagellates by a factor of 10 during the vegetation period [53], and the longitudinal increase in bacteria and heterotrophic flagellates is comparably low [59]. Finally, although our approach cannot resolve the gross turnover rates of individual processes, to estimate the potential contribution of in-stream N cycling to catchment N export, it is essential to use rates of net N retention, i.e., the balance between N uptake and release.

### 4.3. N-Retention Efficiency in Large Rivers

Denitrification is thought to be the main process for N removal from rivers. For the Elbe, previous studies on denitrification support the conclusion that the TN retention of 17 mg N m$^{-2}$ h$^{-1}$ almost entirely equates to denitrified N [13,19] (cf. Section 4.1). Global estimates of denitrification in rivers result in average annual in-stream denitrification rates between 1.5 mg N m$^{-2}$ h$^{-1}$ (estimate based on the SPARROW model calibrated for 30 rivers in eastern U.S. [60]) and 9.6 mg N m$^{-2}$ h$^{-1}$ (average of 41 studies across the U.S., Spain, and Scandinavia [61]), with high variation of individual results integrated in these studies. In comparison, the TN retention observed in the Elbe is relatively high for the following reasons: First, the high $NO_3$–N inputs from the mainly agricultural catchment and the intense production of phytoplankton that serves as labile organic carbon source provide a good substrate supply for denitrification. Furthermore, previous studies along the free-flowing parts of the Elbe showed that the nitrogen (and carbon) turnover can reach up to 1 m into the riverbed [13,29], which stimulates high turnover rates per unit area and time. The majority of empirical studies use laboratory methods that do not integrate the processes occurring in such deep layers of the sediment. Hence, the high biogeochemical activity of the river and methodological differences may cause the comparably high rates found here.

From a river network perspective, several studies suggest a general decrease of the efficiency in N retention with increasing stream order [4,5,9]. The reach-specific TN retention (that is the proportion of the TN input to a reach that is removed within that reach (cf. [9])) is a parameter that mirrors this efficiency. However, this value depends not only on the retention rate but also on the water travel time as well as the amount and the distribution of N inputs along the specific reach. Model-based calculations assuming a uniform direct watershed N-loading in a river (100 kg N km$^{-2}$ watershed y$^{-1}$) resulted in a reach-specific TN removal of 5% for seventh and eighth order rivers [8]. The Elbe is an eighth

order lowland river and, despite its even higher watershed-related TN inputs of 320–470 kg N km$^{-2}$ watershed y$^{-1}$ (calculated on the basis of the TN input of 1361–2030 g N s$^{-1}$ shown in Table 2 and a catchment area at Geesthacht, Elbe-km 586, of 135,000 km$^2$), the reach-specific TN retention in this study averaged 29 ± 10% (Table 2). This is comparably high and therefore contrasts with the hypothesis of a generally low efficiency of TN removal in high-order rivers. Most studies modeling N retention patterns along river networks focus on the influence of hydraulics (e.g., channel morphology and discharge). However, the % nutrient retention is the outcome of both hydraulics as well as biogeochemical reactivity. The factors influencing biogeochemical reactivity, such as NO$_3$ and organic carbon supply, temperature, or hyporheic flow path do change from headwaters to large rivers [7,62] and have been found to be more important for N retention variability than previously thought [63]. Hence, the high biogeochemical reactivity of the Elbe may also explain the comparably high % retention rates found here. These findings underline the problem that model-based conclusions on entire watersheds are simplified and often based on extrapolations using data that mainly derived from smaller streams that do not necessarily represent the conditions in larger rivers. This renders general assumptions about dentrification in large rivers, for which empirical data are lacking, difficult, and researchers should carefully consider the formulations included in their empirical models, especially when taking large rivers into account [64–66].

## 5. Conclusions

Although the different pathways of N retention in aquatic systems are quite well understood, our ability to quantify the actual in situ rates remains incomplete, especially for large rivers. Here, we successfully quantified in situ retention rates for NO$_3$–N and TN along the entire free-flowing German part of the river Elbe by mass balance. Our data add important in situ data of large rivers that can improve future upscaling calculations and calibrations of network-scaled models. With average retention rates of 17 mg N m$^{-2}$ h$^{-1}$ for both NO$_3$–N and TN, our results demonstrate that large, free-flowing rivers can act as considerable TN sinks. In fact, the TN retention measured here is higher than would be expected on the basis of river network models. However, TN concentrations remain high in the downstream reaches of the Elbe and contribute to the eutrophication of estuarine and coastal waters. A better mechanistic understanding of the potential environmental controls of N retention in large rivers is needed to further refine our understanding of N removal along river networks and to decide where and how to act to efficiently reduce N pollution in downstream reaches.

**Author Contributions:** Conceptualization, S.R. and H.F.; Data curation, S.R.; Formal analysis, S.R.; Funding acquisition, H.F.; Methodology, S.R.; Project administration, H.F.; Writing—original draft, S.R.; Writing—review and editing, H.F.

**Funding:** This research was funded as part of the NITROLIMIT project (http://www.nitrolimit.de) in the framework program "Research for Sustainable Development" of the German Federal Ministry of Education and Research (BMBF), Grant numbers 033L041G and 033W015GN.

**Acknowledgments:** We thank Dagmar Steubing, Franz Leiendecker and Carsten Viergutz for their assistance during the field samplings and the laboratory analyses.

**Conflicts of Interest:** The authors declare no conflict of interest.

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
