# Peer review of "A Mass Balance of Nitrogen in a Large Lowland River (Elbe, Germany)"

_water, doi:10.3390/w11112383_

Round 1
Reviewer 1 Report
The authors have elaborated an interesting manuscript trying to deepen into the behavior of N compounds along the Elbe river system. The manuscript is well written and structured and hypotheses and methods are soundly presented. The sampling approach is relevant to the study although the mentioned 1D model limited when considering its precision to accurately identify the time/space location of the reference water volume. Authors should consider discussing this aspect in the manuscript and, if necessary, to provide an assessment about its impact in terms of uncertainty over the sampling performed. The reviewer has a number of questions that would like to submit to the consideration of the authors. They are summarized next: a
a) A perhaps wider context about the Elbe river system and its acknowledged impacts (lignite mining, industrial activities, urban concentrations, etc.) should help to better focus the behavior of this system. b) Although the title clearly identify N as the target system to investigate, it is a pity not to provide information about the behavior of P due to its obvious connections with the studied processes. Some comment justifying why P is left apart from the study should be included in the paper. c) The authors do not explain how they assess the corresponding wetted sediment surface along their transect and the associated uncertainty. d) Have the authors considered atmospheric N deposition (c.f. Beyn et al , 2014) in their model? e) In figure 4, it looks like that there is a significant contrast in terms of zooplankton abundance before and after the connection with the Havel river. Do the authors think that this is the result of the normal evolution of the Elbe river along its course or it may be that reflecting a stronger impact of the Havel over the Elbe river? Do the data obtained support any of the two possibilities?
Ref. Beyn, Mathias and Dänhke (2014) Changes in atmospheric nitrate deposition in Germany- An isotopic perspective. Environ. Pollut. 194, 1-10
Reviewer 2 Report
See attached

Reviewer 3 Report
Ritz and Fischer have made an excellent work with lots of efforts to capture the different pieces of the puzzle that conforms the nitrogen mass balance of a large river. As the authors state there is little research on the significance of large rivers for nitrogen dynamics.
I have several concerns. Firstly, in the introduction, it is unclear whether the goal of the researchers is to study denitrification or nitrogen retention. There are several mentions throughout the introduction to the relevance of denitrification (e.g. line 40) and the difficulties imposed to study it (e.g. line 55). However, in the final section of the introduction the goal of the work is different and refers to the estimation of the of the factors driving the Nitrogen mass balance in the studied reach.
Secondly, in the hypothesis and introduction the role is denitrification is emphasized however there is no actual estimation of the process.
Thirdly, I wonder to which extent observed results are affected by phosphorous concentrations. Where the microorganisms limited in phosphorous or they could benefit from hypertrophic conditions? To which extent could phosphorous influence the findings and explain between-catchment differences?
Finally, concluding remarks regarding their specific goal and hypothesis are needed to provide the reader with a take home message and a final summary of ideas.
Minor comments
L35-36: please specify in which deposition alters the total N pool. Both assimilation and deposition alter the total pool and both reduce, it is unclear why there is a distinction between both.
L36: “by these processes” is redundant with “due to” at the beginning of the sentence.
L38: So far in the introduction there is no reference to different nitrogen compounds, maybe just referring to Nitrogen will be more accurate.
Additionally, reactive nitrogen (line 33) and dissolved nitrogen fraction (line 34) are not exactly the same. Dissolved nitrogen can also include dissolved organic nitrogen. I would clearly state that dissolved fraction refers to the inorganic one. Furthermore, cyanobacteria as nitrogen fixers is not considered as potential source of nitrogen that alters the studied mass balances.
L69-71: As far as I understand, this is the objective of the study, however it is placed a bit in the middle of the text followed by additional discussion and far away from the hypothesis (L88-89). I suggest the authors to discuss the type of sampling and use of hydrodynamic models before stating the goal as part of the selection of the most appropriate study design for large rivers. Furthermore, details about the studied reach are to be provided at the study site section to ease the reading, unless the study site is highly relevant for the introduction.
L139-144: Is there a difference between zooplankton and seston, if yes, how were these two compartments separated?
L214: Please, indicate what POC refers to.
Overall results section: I wonder whether statistical tests will be needed to determine similarities and differences between reaches and time of the year.
L432-434: Would it be possible to add some further details/discussion about the reasons for such differences? Which are potential local drivers that differentiate between-catchment results?
L436-441: While the concluding sentences are very interesting and open up new questions, the discussion seems to miss some closing sentences on the proposed goal. However, if these final sentences are concluding remarks regarding the actual goal, this might need to be clearly stated in the introduction. Furthermore, this is not reflected in the abstract.
Round 2
Reviewer 3 Report
The authors have improved the manuscript. I have however some additional concerns.
The introduction is quite emphatic on denitrification (e.g. lines 52-60, 45-49). However, this seems disconnected from the actual goal of the manuscript (L 71-73). I suggest that the authors focus the introduction on the need to and limitations of quantifying nitrogen retention (of any type) at the ecosystem scale.
L103: I am not sure if reach is the correct term for such a long stretch of river, what about segment following the classification of Frissell et al., 1986.
Frissell, C.A.; Liss, W.J.; Warren, C.E.; Hurley, M.D., 1986: A hierarchical framework for stream habitat classification viewing streams in a watershed context.
L108-111: I wonder, would hyporheic exchange influence the estimations?
L144-146: This sentence is a bit unclear. To what referrers “this increased mass balance”?
Figure 5. Please indicate in the caption what “others” refer to.
L392-400. Though the message of this paragraph is clear, I suggest presenting it in a more positive style. For example, “despite the limitations imposed by the conversion factors used and their temporal dynamics, our results….”. I suggest following a similar approach on the next paragraph (L401-412).
L392-418 read as if the authors were discussing their approach using the model HYDRAX rather than their results and their significance. I suggest organizing the discussion in such a way that the methodological discussion is clearly differentiated from the goal-related discussion.
L454-471. In this paragraph the authors seem to compare results from denitrification estimates with their retention estimates. This comparison might need to be careful, since it denitrification is one piece of retention. Furthermore, the discussion about the contribution of the sediments is a bit disconnected from the rest of the paragraph. Could the authors specify how where the sediments considered in their calculations?
L485-486. Following this idea, I suggest the authors checking the ideas on the Damkohler number. (Oldham et al., 2013)
Oldham, C.E., Farrow, D.E., Peiffer, S., 2013. A generalized Damköhler number for classify- ing material processing in hydrological systems. Hydrol. Earth Syst. Sci. 17, 1133–1148. http://dx.doi.org/10.5194/hess-17-1133-2013.
L496-498. This concluding sentence is a bit unclear, could it be merged with the previous paragraph or added to the conclusion?
Round 3
Reviewer 3 Report
The authors have considered all the comments, I have no further concerns.